# Modeling of Growth and Organic Acid Kinetics and Evolution of the Protein Profile and Amino Acid Content during *Lactiplantibacillus plantarum* ITM21B Fermentation in Liquid Sourdough

**DOI:** 10.3390/foods11233942

**Published:** 2022-12-06

**Authors:** Mariaelena Di Biase, Yvan Le Marc, Anna Rita Bavaro, Stella Lisa Lonigro, Michela Verni, Florence Postollec, Francesca Valerio

**Affiliations:** 1Institute of Sciences of Food Production, National Research Council, Via Amendola 122/O, 70126 Bari, Italy; 2ADRIA Food Technology Institute, UMT ACTIA 19.03 ALTER’iX, ZA Creac’h Gwen, F-29196 Quimper, France; 3Department of Soil, Plant and Food Science, University of Bari Aldo Moro, Via G. Amendola 165/A, 70126 Bari, Italy

**Keywords:** fermented flour, protein profile, growth models, organic acid modeling, in silico simulations

## Abstract

The application of mathematical modeling to study and characterize lactic acid bacterial strains with pro-technological and functional features has gained attention in recent years to solve the problems relevant to the variabilities of the fermentation processes of sourdough. Since the key factors contributing to the sourdough quality are relevant to the starter strain growth and its metabolic activity, in this study, the cardinal growth parameters for *pH*, temperature (*T*), water activity (*a_w_*), and undissociated lactic acid of the sourdough strain *Lactiplantibacillus plantarum* ITM21B, were determined. The strain growth, *pH*, organic acids (lactic, acetic, phenyllactic, and hydroxy-phenyllactic), total free amino acids, and proteins were monitored during fermentation of a liquid sourdough based on wheat flour and gluten (Bio21B) after changing the starting *T*, *pH*, and inoculum load. Results demonstrated that the different fermentation conditions affected the strain growth and metabolite pattern. The organic acid production and growth performance were modeled in Bio21B, and the resulting predictive model allowed us to simulate in silico the strain performances in liquid sourdough under different scenarios. This mathematical predictive approach can be useful to optimize the fermentation conditions needed to obtain the suitable nutritional and technological characteristics of the *L. plantarum* ITM21B liquid sourdough.

## 1. Introduction

The predictive mathematical models applied to bacterial growth and metabolite production can offer significant support to the standardization of fermentation processes, including sourdough, which is obtained by the spontaneous fermentation of a mixture of flour and water operated by lactic acid bacteria (LAB) and naturally occurring yeasts in raw materials [1]. The sourdough fermentation is considered a biotechnological approach that is applied to the bakery product sector to improve the nutritional and functional values of final products when starter strains with known features are used to pilot the fermentation. Among the microorganisms that can be used as starters in sourdough, LAB represent a major group due to their ability to convert the flour constituents into bioactive compounds. LAB are intentionally added to start food fermentations [2], confer functional properties to foods when used as probiotics [3], and prolong sourdough bread shelf-life [4] due to their ability to produce antimicrobial metabolites [5]. Among LAB, *Lactiplantibacillus plantarum* is considered to be the most representative species in sourdough [1]. In particular, the *L. plantarum* ITM21B strain has been used to prolong bread shelf-life due to its production of antimicrobial metabolites, such as lactic, acetic, phenyllactic (PLA), and hydroxy-phenyllactic (OH-PLA) acids [6,7], and its ability to confer nutritional features to salt-reduced yeast-leavened bread via the addition of liquid sourdough enriched in amino acids and organic acids, which compensate for the salt reduction [8,9]. During fermentation, several changes occur in the sourdough, such as the lowering of *pH*, increasing of total titrable acidity (TTA), hydrolysis of flour proteins, production of amino acids, organic acids, and other metabolites responsible for the nutritional and functional values of sourdough-containing bread, and its improved shelf-life [1]. Traditionally, the sourdough process requires long fermentation times (up to 5 days) but recently other approaches have been used to adapt the sourdough process to industrial needs. In our previous studies, the liquid sourdough produced using *L. plantarum* ITM21B as a starter (Bio21B) was obtained after 14 hours of fermentation to meet the industrial needs and was successfully applied to the yeast-leavened bread-making process, leading to a final product with reduced salt content and appreciated sensorial qualities [8,9].

Among the strategies adopted to control and improve the fermentation processes of predictive microbiology, a branch of food microbiology based on mathematical approaches to study the behavior of microorganisms in foods can provide support.

In recent decades, predictive microbiology has been mainly applied to assist food industries in food safety management. The European Union (EU) Regulations and Directives on food safety and hygiene [10] established the importance of risk assessment as a preliminary step to adopt the appropriate control measures, including the use of predictive microbiology. In this approach, mathematical models have been developed to describe the effects of intrinsic and extrinsic factors (intrinsic, such as *pH* and water activity *a_w_*, and extrinsic factors, such as temperature *T*, characterizing the food and its environment) on microbial kinetics and, finally, are used to predict the behaviors of microbial populations in food [11]. For this purpose, several microbial databases, model repositories, and decision tools have been created from microbial data sets, relevant to growth and inactivation kinetic data. Some examples include ComBase (https://www.combase.cc/index.php/en/ accessed on 18 October 2022), Sym’Previus (https://symprevius.eu/en/, accessed on 18 October 2022), MicroHibro (https://www.microhibro.com/, accessed on 18 October 2022), Pathogen Modeling Program (https://pmp.errc.ars.usda.gov/, accessed on 18 October 2022), and others [12]. So far, predictive microbiology has been mainly used to reduce the risk of food contamination by pathogenic or alternative microorganisms, but it can also be applied to predict the growth of technological strains in order to optimize the fermentation processes [13,14]. As an example, Ganzle et al. [15] investigated the effects of pH, temperature, ionic strength, lactate, acetate, and ethanol levels on the growth of three strains representative of the sourdough microflora, to develop a predictive model aimed at identifying the most important factors contributing to the stable association of lactobacilli and yeasts. More recently, Altilia et al. [16] developed models for predicting the growth kinetics of microbial consortia in sourdough. Di Biase et al. [17] developed a cardinal-type growth model to predict the effect of temperature and pH on the maximum specific growth rate of four *L. paracasei* strains, including a probiotic one. The model allowed for simulating and predicting the time and conditions needed to reach the targeted probiotic level in fermented white cabbage [17]. The metabolites produced by LAB during fermentation, such as organic acids, protein, and amino acids, modify the intrinsic characteristics of the medium and, consequently, its nutritional characteristics. Several mathematical models have been developed to determine the kinetics of the production of lactic acid [18,19,20,21], while, to our knowledge, no studies have been performed to model the kinetic production of other organic acids occurring in sourdough, such as PLA and OH-PLA acid associated with the antimicrobial properties of sourdough [6], and to a specific taste of the final product [9]. In this way, predictive mathematical modeling could be a very useful tool for scientists and industries, to optimize food processes and improve the quality of products.

The main aim of the study was to use mathematical modeling of the growth and metabolic kinetics of a sourdough strain and then predict its behavior during fermentation, in order to reduce the high variability of the process mainly related to the starting conditions.

To achieve this goal, the *L. plantarum* ITM21B was chosen as the sourdough starter strain due to its ability to pilot the fermentation process and produce organic acids with a role in the shelf-life extension of sourdough bread [6], and in the taste of salt-reduced bread, together with total free amino acids (TFAA) [9]. The strain was used to develop a predictive model for the effects of *T*, *pH*, *a_w_*, and lactic acid on its growth. In addition, the metabolite production (bioactive organic acids lactic, acetic, PLA, OH-PLA, TFAA, protein content, and protein hydrolysis) of the strain in liquid sourdough (Bio21B) under three starting fermentation conditions was determined. Data collected were then used to simulate in silico the growth of *L. plantarum* ITM21B, the increase in organic acid concentration, and the decrease in pH during fermentation, with the aim of being able to optimize the initial fermentation conditions for achieving the desired strain density, pH, and metabolite profile. This approach can be useful for bakery companies that need to realize standardized and reproducible food products, as well as for scientists who need to deeply study the factors governing the complex process of fermentation.

## 2. Materials and Methods

### 2.1. Bacterial Strain

In this study, we used the *Lactiplantibacillus plantarum* strain ITM21B, named accordingly via a recent re-classification by Zheng et al. [22]; the strain belongs to the Culture Collection of the Institute of Sciences of Food Production, National Research Council (ISPA-CNR). The strain was previously isolated from sourdough [23] and is well known for its preservative [6,7,24] and pro-technological [8,9,25] features. For long-term storage, stock cultures were prepared by mixing 8 mL of a culture in de Man Rogosa Sharpe (MRS) broth (Biolife Italiana S.r.l., Milan, Italy) with 2 mL of Bacto glycerol (Difco, Becton Dickinson, Co., Sparks, MD, USA) and freezing 1 mL portions of this mixture at −80 °C. To obtain fresh cultures, the strain was subcultured twice (1% *v*/*v*) in MRS broth for 24 h at 37 °C before use in experiments. 

### 2.2. Growth Kinetic of L. plantarum in Broth

Growth parameters for *T*, *pH*, *a_w_*, and undissociated lactic acid (e.g., minimal, optimal, and maximal values of temperature for growth, and minimum inhibitory concentration of undissociated acid) were determined from experiments performed in broth. Each factor was tested in a mono-factorial design except for lactic acid, whose effect was evaluated at two pH levels, such as in Coroller et al. [21]. Growth kinetics were performed in 0.22 mm-filtered modified de Man Rogosa and Sharpe (MRS) broth lacking acetic acid and citrate (pH 6.2), according to Cuppers and Smelt [26]. The pH value of the medium was adjusted using 5N HCl or 5N NaOH to the modified MRS and recorded by the pH meter (Beckman Coulter, model 340, supplied with a glass electrode Beckman Coulter, Brea, CA, USA); the influence of *a_w_* was studied using different percentages of NaCl (a_w_/[NaCl]) recording values by the AquaLab Series 3 (Decagon Devices, Pullman, Washington, DC, USA); two preset pHs (4.7 and 5.1) of the medium added with DL-lactic acid 85% (Sigma-Aldrich, Inc. St. Louis, MO. USA) were used. For the *L. plantarum* strain, the maximum specific growth rate (µ*_max_*) was determined for about eight temperatures (from 5.5 to 39 °C), sixteen pH values (from 3.2 to 9.1), eight percentages of NaCl (from 0 to 6.5% *w*/*v*) corresponding to *a_w_* values ranging from 0.962 to 0.996, and nine concentrations (from 0 to 100 mM) of DL-lactic acid 85% in its undissociated form [*HA*] at pH 4.7 and 5.1. The *pH*, *a_w_* and lactic acid experiments were performed below the optimum temperature for growth (*T_opt_*) [27]. The experiments were conducted at 30 °C for *pH* and *a_w_* and at 25 °C for lactic acid. For studied temperatures ranging from 18 to 39 °C, the growth of the strain was automatically monitored by a Bioscreen C (Labsystems, Helsinki, Finland) using the turbidimetry method [28] according to the dilution methods reported by Cuppers and Smelt [26]. A total of 20 controls (2 technical replicates per experimental condition) were performed for each plate reader. Optical density (OD) measurements at 600 nm were taken at 27 min intervals. For each condition, two technical replicates and two biological replicates were performed. The recorded temperature of the Bioscreen C chamber was stable (±0.1 °C) at each set temperature. When growth kinetics could not be automatically recorded by the Bioscreen C, they were determined manually after static incubation in modified MRS at the tested temperatures (5.5 and 11 °C) using the same culture procedures of the inoculum preparation. Briefly, 50 mL of modified MRS broth were inoculated with initial cell concentrations of 10^3^ cfu/mL for each experimental condition. The recorded temperature of the incubators, monitored by using an electronic HOBO Temp/RH data logger (Onset Corp., Bourne, MA, USA), was stable (±0.1 °C) at each set temperature. At appropriate time intervals, aliquots of the cultures were decimally diluted in sterile NaCl (0.85% *w*/*v*) + Tween 80 (0.025%) and 100 μL of each dilution was spread on MRS agar plates incubated for 48 h at 37 °C.

### 2.3. Growth Kinetic of L. plantarum in Liquid Sourdough (Bio21B) Samples

The experimental protocols and data analysis were performed according to standardized and reported methods in order to ensure reliable simulations of growth in the matrix [29,30]. The *L. plantarum* ITM21B strain was used to inoculate the liquid sourdoughs as reported by Di Biase et al. [8]. Briefly, Bio21B-1, -2, and -3 were prepared by mixing 320 mL of sterile water, 40 g of wheat flour, and 40 g of wheat gluten, obtaining a dough yield (DY, dough weight × 100/flour weight) of 500. Cells of the ITM21B strain from 24 h culture in MRS broth were washed twice with sterile distilled water, inoculated in the mixture at a suitable final cell density (log cfu/mL), and incubated for 28 h at different starting fermentation conditions (Table 1). The pH value of Bio21B-3 was reached, adding DL-lactic acid 85% (Sigma-Aldrich, Inc. St. Louis, MO. USA) as the acidifier, since it is a food-grade compound and can be used in a food company. The growth parameters of *L. plantarum* ITM21B in the liquid sourdough Bio21B at different *pH*, temperature, and starting strain load values were determined. For each condition, two biological replicates were performed.

### 2.4. Microbiological and Physicochemical Analyses of Bio21B during L. plantarum Fermentation

At different time intervals, samples were collected for microbiological analysis, determination of pH, TTA, organic acids (lactic, acetic, PLA and OH-PLA), TFAA, total protein degradation, and protein pattern. Experiments were carried out twice and analyzed in duplicate (2 × 2).

Serial decimal dilutions of Bio21B in sterile NaCl (0.85% *w*/*v*) + Tween 80 (0.025%), were prepared, and a 100 µL aliquot of each dilution was spread on MRS plates. which were incubated for 48 h at 37 °C. The total LAB count was expressed as log cfu/g. To monitor the presence of the *L. plantarum* ITM21B strain inoculated in Bio21B, at each sampling time, 20% of total colonies randomly selected from countable MRS agar plates were isolated and checked for purity, as reported by Di Biase et al. [8].

Bacterial DNA from presumptive isolated LABs colonies was extracted from overnight cultures grown in MRS broth at 37 °C and analyzed by rep-PCR using the primer pair REP-1R-Dt/REP-2R-Dt, as previously described by De Bellis et al. [31]. The identification of *L. plantarum* ITM21B was based on the comparison of the specific pattern obtained from pure cultures of *L. plantarum* ITM21B. The amplification products were separated by the Lab-on-a-Chip (LoaC) capillary electrophoresis using the DNA7500 LabChip Kit and the DNA LabChip platform (Agilent Technologies, Waldbronn, Germany). DNA 7500 ladder was used as the size standard and as a normalization reference, in addition to lower (50 bp) and upper (10.380 bp) markers added to each DNA sample on the chip. The reproducibility of the fingerprints was verified by repeating the analysis twice.

The pH of Bio21B was recorded at each sampling time with a portable pH meter (type110, Eutech Instruments, Ayer Rajah Crescent, Singapore) supplied with the Double Pore D electrode (Hamilton, Bonaduz, Switzerland). TTA was measured according to AOAC method no. 981.12 [32] and expressed in mL of 0.1 N NaOH required to achieve a pH of 8.3.

### 2.5. Determination of Organic Acids in Bio21B

Sample preparation and analysis of lactic, acetic, PLA, and OH-PLA acids were performed as reported by Di Biase et al. [8]. Briefly, 10 g portions of each Bio21B were diluted in sterile tap water (90 mL), homogenized in a Stomacher (Seward, London, United Kingdom) for 2 min, then the suspensions were centrifuged (9072× *g*, 10 min, 4 °C) and the supernatants were freeze-dried. The freeze-dried samples were resuspended in the HPLC mobile phase (0.007 M H_2_SO_4_) (Fluka, Deisenhofen, Germany) and filtered by centrifugation (7000× *g*, 1 h, 2 °C) through a 3000 Da cut-off microconcentrator (Ultracel-3k, Amicon, Danvers, MA, USA). The fraction containing molecules with molecular weight lower than 3000 Da was analyzed by HPLC (AKTABasic10 system, Amersham Pharmacia Biotech Europe GmbH, Freiburg, Germany) including a pump P-900, a degasser (Gastorr BG-12, FLOM Corporation, Tokyo, Japan), using a Rezex ROA organic acid H^+^ (8%) column (7.80 mm × 300 mm, Phenomenex, Torrance, CA, USA), an injection volume of 10 µL, a 3-channel UV detector (Amersham Pharmacia Biotech Europe GmbH) set at 210 (lactic, acetic, PLA acids) and 220 nm (OH-PLA). The mobile phase was pumped at a flow rate of 0.7 mL/min through the column heated to 65 °C. Quantification of the organic acids was performed by integrating calibration curves obtained from the relevant standards. The limit of detection (LOD) and limit of quantification (LOQ) were calculated considering a signal-to-noise ratio (S/N) of 3 and 6, respectively. LOD values were the following: lactic acid, 0.263 mM/kg; acetic acid, 0.279 mM/kg; PLA, 1.08 μM/kg; OH-PLA, 0.774 μM/kg. LOQ values corresponded to 2 × LOD. The final concentration of each organic acid in Bio21B samples was calculated considering the concentration and/or dilution factors and expressed as mM/kg or µM/kg of the product.

### 2.6. Total Free Amino Acids, Protein Content, and Profile of Bio21B

Water/salt-soluble extracts of the Bio21B samples were prepared according to the method originally described by Osborne [33] and modified by Weiss et al. [34] and used to analyze total free amino acids (TFAAs). TFAAs were determined by the Cd-ninhydrin method, as reported by Doi et al. [35].

Total proteins were extracted from Bio21B samples (during fermentation time) under reducing conditions. Briefly, 40 mg of flour—or an equivalent amount of Bio21B samples weighed on the basis of their flour content—were mixed with 200 µL of an extraction solution containing 5% mercaptoethanol and 2% SDS [36] for 3 h at 4 °C. Afterward, insoluble material was removed by centrifugation (12,000× *g* for 15 min), and supernatants, after heating at 100 °C for 2 min, were stored at −20 °C and used for electrophoresis. The protein concentration was determined by the Bradford method [37] using the Bio-Rad dye reagent (Bio-Rad Laboratories, Hercules, CA, USA) with bovine serum albumin as the standard and expressed as mg of protein per gram of flour. The total protein degradation (*TPD*) during fermentation was expressed as a percentage and quantified using the following equation:(1)$$\%\;TPD=100-\left[\left(\frac{TP_{f}}{TP_{i}}\right)\times 100\right]$$
where *TP_f_* represents the total protein content after fermentation and *TP_i_* is the initial total protein content (before fermentation). Total protein extracts were analyzed by the LoaC capillary electrophoresis using the Protein 230 Lab Chip kit (Agilent Technologies, Waldbronn, Germany) with a molecular weight range of 14–230 kDa. Sample preparation and chip loading was performed according to the manufacturer’s instructions and the data evaluation was carried out by the dedicated 2100 Expert software that aligns sample proteins to a molecular weight ladder using internal standards. The software displays automatic data, referring to the samples as peaks (electropherogram) and bands (gel-like image) in a tabular format that reports each protein peak, molecular weight (Mw), time-corrected peak area (TCA), relative concentration (RC) based on a one-point calibration to the upper marker (60 ng/mL), and protein percentage (%) calculated on the total peak areas in the sample. For each molecular weight area, the sum of these percentages was calculated. The manual integration of peaks was performed after each run and peaks with RC < 20 ng/mL were excluded from the analysis as the significance was low considering the detection limit of the method. All experiments were performed twice (n = 2).

### 2.7. Mathematical Modeling

#### 2.7.1. Determination of the Maximum Specific Growth Rates

For the Bioscreen experiments, the maximum specific growth rate (µ*_max_*) values were directly derived from the slope of the linear relation between time to detection (TTD) and the logarithm of the inoculum size [26]. The TTD was defined as the time at which a certain well reached a specific OD600 value of 0.15. In the manual experiments (either in broth or in Bio21B), the maximum specific growth rate (µ*_max_*) was calculated by fitting the growth curves to the logistic model with delay and rupture [38]. Secondary model fittings were performed with a non-linear fitting module (NLINFIT, MATLAB, R2019b, The MathWorks, Natick, MA, USA). Before fitting, a usual square root transformation was performed to homogenize the variance of the maximum specific growth rate. Simulations of bacterial growth were performed using an in-house program developed in the MATLAB language.

#### 2.7.2. Temperature, *pH*, *a_w_*, and Lactic Acid Models

A multiplicative model (Equation (2)) was used to relate µ*_max_* to temperature, *pH*, *a_w_*, and undissociated lactic acid [*HA*] (mM):(2)$${µ}_{max}\left(T,\;pH,\;a_{w},\;\left[HA\right]\right)={µ}_{opt,\;MRS}\;\tau \left(T\right)\gamma \left(pH\right)\rho \left(a_{w}\right)\delta \left(\left[HA\right]\right)$$
where *τ* (*T*), *γ* (*pH*), *ρ* (*a_w_*), and *δ* ([*HA*]) are the normalized effects of *T*, *pH*, *a_w_*, and concentration of [*HA*], µ*_opt_,_MRS_ is* the optimum maximum specific rate (µ*_max_*) in modified MRS.

For the effect of temperature, we used the cardinal temperature model with inflection (CTMI) [39]:(3)$$\tau \left(T\right)=\frac{\left(T-T_{max}\right){\left(T-T_{min}\right)}^{2}}{\;\left(T-T_{min}\right)\left(T_{opt}-T_{min}\right)\left(T-T_{opt}\right)-\left(T_{opt}-T_{max}\right)\left(T_{opt}+T_{min}-2T\right)}$$
where *T_min_*, *T_opt_*, and *T_max_* are the minimum, optimum, and maximum *T* for growth.

Most pH models assume a close-to-linear evolution of the growth rate as a function of suboptimal (and supraoptimal) pH. As this was not in accordance with our experimental observations, we used the following term reported by Di Biase et al. [17], based on Presser et al. [40,41]:(4)$$\gamma \left(pH\right)=\frac{\left(1-10^{\left(pH_{min}-pH\right)}\right)\left(1-10^{Q\left(pH-pH_{max}\right)}\right)}{\left(1-10^{\left(pH_{min}-pH_{ref}\right)}\right)\left(1-10^{Q\left(pH_{ref}-pH_{max}\right)}\right)}$$
where *pH_min_* and *pH_max_* are the minimum and maximum *pH* for growth, respectively, and *Q* is a shape parameter. The parameter *pH_ref_* was set to 6.2 (i.e., the pH of the modified MRS medium). Equation (4) describes a linear evolution of µ*_max_* as a function of proton concentration in the suboptimal pH range [40]. The parameter *Q* reflects the shape of the relationship pH-µ*_max_* in the supraoptimal range [41]. A value of *Q* equal to 1 would correspond to a linear evolution of µ*_max_* as a function of hydroxide ion concentration. Note also that *γ* (*pH*) is equal to 1 at the pH of the modified MRS medium. This model was successfully used by Di Biase et al. [17] to describe the effect of pH on the maximum growth rate of *Lacticaseibacillus paracasei* strains.

For a_w_/[NaCl], we used a cardinal *a_w_* model with a shape parameter of 1, as in Couvert et al. [42]:(5)$$\rho \left(a_{w}\right)=\frac{\left(a_{w}-1\right)\left(a_{w}-a_{w,min}\right)}{\;\left(a_{w}-a_{w,min}\right)\left(a_{w}-1\right)-{\left(a_{w}-a_{w,opt}\right)}^{2}}$$
where *a_w_,_min_* and *a_w_,_opt_* are the minimum and optimum *a_w_* for growth, respectively.

In Equation (5), *a_w_,_opt_* was set to 0.997.

The [*HA*] term is as follows [21]:(6)$$\delta \left(\left[HA\right]\right)=1-{\left(\frac{\left[Lactic\;acid\right]}{\left(1+10^{\left(pH-3.86\right)}\right)MIC_{U}}\right)}^{\alpha }$$
where *MIC_U_* is the minimum inhibitory concentration of undissociated lactic acid.

Based on the experimental data, the shape parameter α was set to 1 as in Le Marc et al. [43]. Equation (6), therefore, describes a linear relation between µ*_max_* and the concentration of undissociated lactic acid. A common value of 3.86 was used for the pKa of lactic acid.

The *lag* time was taken into account through the parameter *h_0_* (also called the “work to be done” before the bacterial growth could start, Robinson et al. [44]):(7)$$lag=\frac{h_{0}}{{µ}_{max}}$$

#### 2.7.3. Effects of *L. plantarum* Growth on the Nutritional Profile

The production of organic acids by *L. plantarum* was described by this common equation (see e.g., [18,19]):(8)$$\frac{dP}{dt}=Y_{p}\;\frac{dN}{dt}+m_{P}N$$
where *P* is the concentration of the organic acid, *Y_p_* (mM/kg/cell or µM/kg/cell) is the organic acid concentration produced by cellular division by time unit and *m_p_* (mM/kg/cell.h or µM/kg/cell.h) is the organic acid concentration produced by a cell by time unit.

First, the growth curves of *L. plantarum* were fitted with the model of Baranyi and Roberts [45] using non-linear regression. Then, the growth-associated coefficient (*Y_p_*), and non-growth-associated coefficient (*m_p_*) were estimated from product formation models by minimizing the sum of the squared errors between the observed and simulated concentrations of the organic acid. Note that the parameter *m_p_* was set to 0 for OH-PLA as its production was only observed during the growth phase (and not the stationary phase). The system of equations for bacterial growth and end-product formation was solved numerically by the Runge–Kutta method (ODE23, MATLAB R2019b, The MathWorks, Portola Valley, CA, USA). The estimation of the model parameters was performed using a non-linear fitting module (NLINFIT, MATLAB R2019b, The MathWorks).

#### 2.7.4. Determination of Kinetic Growth Parameters of *L. plantarum* ITM21B in Bio21B for the In Silico Simulations

The growth curve of *L. plantarum* ITM21B in Bio21B-2 was used to determine the optimal growth rate (µ*_opt,Bio21B_*) and *h*_0_ in sourdough Bio21B, following the approach described in Pinon et al. [30]. Briefly, µ*_opt,Bio21B_* and *h*_0_ were derived from the µ*_max_* and *lag* values estimated for Bio21B-2 (noted µ*_maxBio21-2_* and *lag_Bio21-2_*):(9)$${µ}_{opt,\;Bio21B}=\frac{{µ}_{maxBio21B-2}}{\tau \left(T\right)\gamma \left(pH\right)\rho \left(a_{w}\right)\delta \left(\left[HA\right]\right)}$$
where *τ* (*T*), *γ* (*pH*), *ρ* (*a_w_*), and *δ* ([*HA*]) are the normalized effects of environmental factors corresponding to the strain growth in Bio21B-2. The µ*_opt_,_Bio21B_* was then substituted to µ*_opt_,_MRS_* in Equation (2) for the simulation of ITM21B growth in Bio21B:(10)$$h_{0}={µ}_{maxBio21B-2}\;lag_{Bio21B-2}$$

Bacterial growth and organic acid formation was simulated at different initial conditions of fermentation (numerically) by the equations for growth and product formation. The values of µ*_opt,Bio21B_* and *h_0_* in Bio21B were also used as inputs. The cessation of growth of *L. plantarum* was assumed to occur when the minimum inhibitory concentration of undissociated acid was reached. The kinetics of pH were deduced from the lactic acid concentration (major acid in Bio21B). The relationship between the pH and the concentration of lactic acid in Bio21B was established using the data collected in the liquid sourdoughs Bio21B-1, -2, and -3.

### 2.8. Statistical Analysis

Data relevant to the starting fermentation conditions (Table 1) within each column (log(N_0_), *T*, *pH*, lactic acid) and data from the amino acid, protein, and organic acid evolution during fermentation were subjected to one-way ANOVA; pair-comparison of treatment means was achieved by Tukey’s procedure at *p* < 0.05, using the statistical software Statistica 10.0 (StatSoft Inc., Tulsa, OK, USA). Data below the LOD were substituted with half of the LOD [9]. The analysis of variance (ANOVA) was performed to detect significant differences within different samples or within fermentation times.

## 3. Results and Discussion

### 3.1. Growth Model Parameters of L. plantarum ITM21B in Liquid Medium

Equation (2) was used to fit µ*_max_* data for *L. plantarum* strain as a function of temperature (*T*), *pH*, *a_w_*, and [*HA*] (Appendix A). The estimated growth model parameters are reported in Table 2. Additionally, Figure 1 shows the good correspondence between the observed and fitted growth rates for the effects of temperature, *pH*, *a_w_*, and lactic acid concentration on the ITM21B growth. The parameters show that this strain can grow in a wide range of temperatures and pH values while the water activity could represent a limiting factor for its growth.

The maximum growth rate was found to increase sharply from *pH_min_* to about pH 4, followed by a plateau between pH 4 and 6, followed by a gradual decrease towards *pH_max_*. The estimated values for *T_min_* (2.40 °C) and *pH_min_* (3.14) for the strain ITM21B are close to the lower range of values observed by Aryani et al. [46] for 19 strains of *L. plantarum* (*T_min_* ranging between 3.4 and 8.3 °C and *pH_min_* values between 3.17 and 3.50). The estimated value for the shape parameter *Q* (Table 2) indicates an absence of a linear relationship between the maximum growth rate and the hydroxide ion concentration in the supraoptimal pH range. With regard to *a_wmin_*, the values estimated in Aryani et al. [46] were slightly lower (0.936–0.953) than that of ITM21B, confirming the high strain variability of growth parameters for this species. Finally, during fermentation, the ITM21B strain growth was inhibited over a concentration of 14.8 mM of the undissociated form of lactic acid, and we can assume that the *MIC_U_* value was not affected by pH in the range of 4.7 to 5.1. The *MIC_U_* estimated in this study (14.8 mM) is, however, lower than the lower and upper limits (29 and 38 mM, respectively) reported by Aryani et al. [46]. This different result can be due to the specific behavior of the strain ITM21B or to the pH of the medium, the incubation temperature, the water activity values, or the difference between the different experimental protocols used.

### 3.2. Growth Performances of L. plantarum ITM21B in Bio21B during Fermentation and Evolution of Metabolites Produced

#### 3.2.1. Microbiological and Physicochemical Parameters Evolution

The fermentation experiments in Bio21B demonstrated the presence of strain *L. plantarum* ITM21B, which represented the totality of LAB isolates, and its growth in liquid sourdough as ascertained by the REP-PCR analysis performed at each sampling.

The kinetic data relevant to the strain growth in Bio21B samples are reported in Table 3. The lower µ*_max_* was registered in Bio21B-1 even if a shorter *lag* time was observed with respect to the other samples. Starting the fermentation with lower pH and inoculum loads (Bio21B-3), the strain reached higher N_max_ although with a longer *lag* time.

As observed in Figure 2 and Figure 3, during fermentation, the strain grew, reaching levels of 8–9 log cfu/g and pH levels < 4. The TTA values ranged between 0.65 and 5 mL NaOH 0.1 N for Bio21B-1, 2.5 and 17 mL for Bio21B-2, and between 3 and 10 mL in Bio21B-3 within 24 h of growth. In our previous studies, a fermentation time of about 14 h was chosen to meet the industrial needs since long sourdough fermentation times are not compatible with the large-scale production of bakery products [8,9,25,47,48]. When the liquid sourdoughs were included in the formulation of yeast-leavened bakery products, the resulting sensorial and technological attributes were comparable to those obtained in traditional sourdough bread. Therefore, in the current study, this fermentation time (T14 h) is used as a reference. In Bio21B-1, 8 log cfu/g, pH lower than 4, and TTA of 2.55, were registered after 14 h of incubation. In Bio21B-2 the TTA value reached levels of about 6 mL NaOH 0.1N at 14 h fermentation, pH of 4.5, and counts of 8.6 log cfu/g. In Bio21B-3, the strain ITM21B reached the count of 8.8 log cfu/g, pH of 4.75, and TTA of 4.5 at 14 h growth. The higher temperature of Bio21B-2 (37 °C), with respect to Bio21B-1, allowed it to reach 8 log cfu/g at 8 h incubation. Using a lower inoculum load (5 log cfu/g), the bacterial proliferation reached values of 8 log cfu/g at 14 h incubation (Bio21B-2). Finally, the *a_w_* was monitored throughout the fermentation without change but it maintained values ranging from 0.997 to 1 in all of the liquid sourdoughs.

#### 3.2.2. Organic Acid Content Evolution

During sourdough fermentation, several metabolites, including organic acids, carbon dioxide, diacetyl, peptides, and alcohols are produced by the microorganisms occurring or intentionally added as starters [49]. These metabolites have different beneficial attributes contributing to improving the sensorial, nutritional, and microbial qualities of final products. In particular, the organic acids and amino acids can have a role in the sensorial quality of bread containing sourdough in its formulation, as demonstrated in our previous studies [8,9]. The results of the research evidenced the suitability of a wheat/gluten liquid sourdough (DY 500, inoculum 8 log cfu/g, 30 °C, pH 6) based on the growth of *L. plantarum* ITM21B and similar to that produced in the current study, to be included in the bread formulation, and realize the yeast-leavened product with reduced salt content due to the presence of metabolites as organic acids, proteins, and amino acids, with a potential role in compensating the negative perception of salt reduction. In particular, 8 log cfu/g and pH of about 3.6, were associated with a fermentation quotient of about 19 (representing the molar ratio between lactic and acetic acids) and a content of 25 and 12 μM/kg of PLA and OH-PLA, respectively. In the current study, the different fermentation conditions (Table 1) significantly influenced the parameters measured and the time needed to reach a pH of 3.6 and/or the presence of the metabolite pattern as registered in our previous studies.

Regarding the production of organic acids (Figure 2), lactic acid was produced at levels of about 50 mM/kg in Bio21B-2 and -3 while a lower concentration (30 mM/kg) was detected in Bio21B-1 at the end of fermentation. Acetic acid, PLA, and OH-PLA were scarcely produced in all liquid sourdoughs. In all Bio21B samples, a pH value of about 3.6 was reached after about 22 h fermentation with a corresponding lactic acid concentration ranging from 25 and 38 mM/kg, thus confirming a relationship between pH and lactic acid concentration. This pH level corresponded to PLA and OH-PLA concentrations higher (between 43 and 58 μM/kg for PLA and between 19 and 21 μM/kg for OH-PLA) than those registered by Di Biase et al. [8], indicating that the production of these acids is more related to the fermentation time. In fact, PLA and OH-PLA started at about 8–10 h for Bio21B-1 and -2 while a later production time (18 h) was observed in Bio21B-3. The acetic acid was detected at levels even lower than 2 mM/kg in all Bio21B samples.

Other authors investigated the PLA production in sourdough. As an example, Vermeulen et al. [50] found a significant improvement in PLA production by an *L. plantarum* strain after the addition of α-ketoglutarate to the sourdough, registering values higher than 1 mM after 48 h incubation. The addition of specific precursors could be an additional strategy to optimize the fermentation process and improve the sourdough quality.

#### 3.2.3. Protein Content Evolution

Protein hydrolysis and amino acid metabolism contribute to the beneficial effects of sourdough fermentation on bread quality [51,52,53]; during fermentation, the protein and amino acid contents considerably change, also contributing to better digestible bakery products [8,54].

During fermentation, the protein and amino acid contents considerably changed. Figure 3 reports the content of total proteins extracted under reducing conditions and quantified by the Bradford assay during Bio21B fermentation. The protein degradation mainly started after 16 h of fermentation, as suggested by the TPD values, which ranged between 62 and 75%.

In Bio21B-1, the total protein content slightly (*p* > 0.05) increased after 10 h from 24.7 ± 4.7 to 27.2 ± 1.0 mg/g and then progressively decreased (*p* < 0.05) to reach a concentration of 1.5 ± 0.2 mg/g at 24 h (TPD = 93.7 ± 1.4%). Similar trends were observed in Bio21B-2 and Bio21B-3, although higher protein contents were found at fermentation times T0h (40.5 ± 0.6 and 47.5 ± 4.7 mg/g, respectively) and T10h (47.7 ± 3.3 and 72.6 ± 8.3 mg/g, respectively); after 24 h of fermentation TPD values of 90.8 ± 1.2% and 86.6 ± 1.8% occurred in Bio21B-2 and Bio21B-3, respectively.

Modifications in protein profiles, explored using LoaC capillary electrophoresis after 1, 14, and 24 h of fermentation, are shown in Figure 4. According to Mw, three different areas were considered in the electropherograms: A1 (14–30 kDa), A2 (31–79 kDa), and A3 (80–230 kDa), and percentages of peaks for each Mw area were gained. At the beginning of fermentation, about 18 protein bands, ranging from 14 to 215 kDa, were observed in all Bio21B products, with highly predominant bands/peaks in the 31–79 kDa range (A2) accounting for 62.7 ± 4.4% of total protein content; minor protein contents were found in A1 (11.1 ± 1.9%) and A3 (26.2 ± 5.8%). After 14 h of fermentations, with respect to fermentation time T0h, all Bio21B samples showed superimposable protein patterns. No changes in the protein percentage distribution occurred in Bio21B-2, while in Bio21B-1 and Bio21B-3, high molecular weight proteins in A3 were degraded (15.6 and 10.8%, respectively) and a concomitant increase of smaller proteins in A1 (18.4 and 17%, respectively) was observed. At T24h, a substantial change in the protein patterns occurred in all Bio21B samples: in particular, the high molecular weight proteins in the A3 region were completely degraded in all Bio21B samples and about 7 protein bands were observed in the 14–58 kDa range with 2 main bands of about 45 and 55 kDa, as also reported by Valerio et al. [9] (Figure 4). At this time, the pH value was significantly lower than 14 h and the acidification favored the proteolysis.

Results confirmed the proteolytic activity of the *L. plantarum* ITM21B strain, which was responsible for characteristic sensorial attributes in bakery goods, as demonstrated by Di Biase et al. [8]. In all Bio21B samples, an almost total protein degradation was observed at the end of fermentation (24–28 h) with a substantial change in the protein pattern and a concomitant lowering of pH. This result was in accordance with Zotta et al. [55], which observed that the major effects in protein hydrolysis were attributable to the pH conditions and proteolytic activity of wheat flour enzymes. Results confirm that proteolysis occurring during fermentation leads to the progressive degradation of high molecular weight proteins, as also observed by Zotta et al. [55], into smaller proteins, and probably also into soluble peptides and free amino acids useful to balance the negative perception of salt reduction along with other molecules.

#### 3.2.4. Amino Acid Content Evolution

As for protein content, although not linear, an increase in the concentration of TFAAs was observed during fermentation (Figure 5). The highest values, up to 351 mg/kg of dough, were found for Bio21B-2 after 27 h of incubation, followed by Bio21B-3 at 28 h with 344 mg/kg. Despite Bio21B-1 showing almost twice the TFAA content, with respect to Bio21B-2 and Bio21B-3 at the beginning of fermentation, it was the liquid sourdough with the lowest increments (roughly 50%) in the first 24 h of incubation. We can assume the conversion of some amino acids into organic acids as it occurs for PLA, which is produced by phenylalanine degradation and OH-PLA from tyrosine, as reported by Valerio et al. [56].

### 3.3. Modeling Results

As determined by the growth curve of strain ITM21B in Bio21B-2, the µ*_opt,Bio21B_*, and *h_0,Bio21B_* were estimated as 1.01 h^−1^ and 3.65, respectively. These values were used to perform the simulations of *L. plantarum* growth in fermentation conditions different from those experimentally tested.

Data relevant to the acid production originating from the strain growth in the liquid sourdoughs (Bio21B-1, 2, and 3) were used to identify the parameters taking into account the transition from the exponential phase (*Y_p_*) to the stationary phase (*m_p_*), i.e., the acidification of the medium because of the production of lactic acid, PLA, and OH-PLA. The estimated parameters for each experimental condition are shown in Table 4. The comparisons between the fitted kinetics of these metabolites and experimental measurements are proposed in Figure 2.

In all Bio21B samples, the production of lactic acid started during the exponential growth phase and continued at a lower production rate during the stationary phase, leading to an increase the molar concentration (Table 4 and Figure 2). Looking at the modeling results, for PLA, the production rate was lower during the growth stationary phase compared with the exponential phase. Note that for each condition, the production of OH-PLA was observed only during the growth phase. Lowering the initial pH to a value of 5, the PLA production rate decreased both in the exponential growth phase and in the stationary one, while for lactic acid only, the parameter *m_p_* was affected.

In relation to lactic acid production, a close relationship with the pH was observed (Figure 6), thus confirming data from other authors [20]. The proton formation (assumed as toxic products) and, consequently, the pH reduction, and some multiple factors, such as substrate depletion, suboptimal temperature, water activity, and other microbial antagonists, can influence the growth rate and cause an induced early stationary phase corresponding to the highest cell number (N_max_). Consequently, the total lactic acid concentration needs to take into account the microbial cell density, the dissociation kinetics of lactic acid in the medium, and the MIC value [20]. The use of the pH measurements could be equivalent to the use of lactic acid concentrations as previously reported [19,20] since, for a given medium, there is a 1–1 relation between the lactic acid concentration and the pH. Figure 6 shows that this relationship was maintained for the three different conditions, although temperature, inoculum load, and initial pH value were different.

### 3.4. Simulation

In order to predict the fermentation time and conditions allowing the growth of *L. plantarum* in a flour-based liquid sourdough until reaching a pH of about 3.6 and a lactic acid concentration of about 21 mM as that obtained in Di Biase et al. [8], different starting conditions of fermentation were used to simulate the growth and acid production in Bio21B (Figure 7). PLA and OH-PLA productions were also modeled and simulated. As observed in Figure 7, scenario A, a high inoculum load (8 log cfu/g) could allow reaching the desired pH of about 3.6 after 14 h fermentation with a concomitant lactic acid production of about 40 mM/kg and PLA and OH-PLA concentrations of about 58 and 16 μM/kg, respectively. A slight discrepancy between data from the same formulation and fermentation conditions used in Di Biase et al. [8] (lactic acid: 21 mM/kg, PLA: 25 μM/kg and OH-PLA: 12 μM/kg) could be due to differences in the raw materials used to prepare the liquid sourdough. The other scenarios demonstrate that the starting fermentation conditions can modify the time to reach pH 3.6 and the relevant lactic acid concentration. Increasing the temperature to 37 °C and lowering the inoculum load (5 log cfu/g) (scenario B), the fermentation time should increase to 17 h and the relevant lactic acid concentration should be about 27 mM/kg, while a significant increase of the PLA and OH-PLA production could be observed (94 and 37 μM/kg). Using a temperature of 30 °C and an inoculum load of 5 log cfu/g (scenario C), the parameters should not significantly change with respect to scenario B, but could be different from scenario A (inoculum load 8 log cfu/g). A further lowering of the inoculum load to 4 log cfu/g, although maintaining the fermentation temperature to 37 °C (scenario D), close to the optimal growth value, could determine a longer time to reach the pH 3.6 (21 h), obtaining almost the same acid concentrations of scenarios B and C (36 mM/kg lactic acid, 99 and 38 μM/kg PLA and OH-PLA). Therefore, data suggest that the inoculum load affects the acid production more than the fermentation temperature. Finally, the target pH could be reached after about 20 h if the starting pH is 5, the inoculum is 8 log cfu/g, and the temperature is 30 °C (scenario E). Interestingly, in this condition, the PLA and OH-PLA concentration should be lower than the other conditions (29 and 9.5 μM/kg) while the lactic acid should reach the value of 31.5 mM/kg. These simulation results indicate that by lowering the starting pH to 5, lower acid production could be expected.

## 4. Conclusions

Due to the need to obtain practical use of sourdough and a more sustainable process at the industrial level, the fermentation time and variability of the process should be controlled, while maintaining the suitable nutritional and technological characteristics of the final product.

In the current study, a predictive model for the effects of temperature (*T*), *pH*, water activity (*a_w_*), and undissociated lactic acid [*HA*] on the growth of a pro-technological *L. plantarum* strain (ITM21B) was developed. The strain is known for its fermentative ability in sourdough. The metabolic features are mainly related to the production of bioactive compounds, such as lactic acid, PLA, OH-PLA, and TFAA; the strain is also known for its suitability in the production of salt-reduced yeast-leavened bread. For this reason, the organic acid production (lactic, PLA, and OH-PLA acids) and growth performance were modeled in liquid sourdough and the resulting predictive model allowed obtaining in silico simulations of the *L. plantarum* ITM21B fermentation performances in Bio21B under different scenarios.

Experimental data demonstrated that the different fermentation conditions significantly affected the strain`s growth and metabolite pattern. Organic acids were mainly produced after the exponential growth phase, while proteins and TFAAs mainly increased at the end of fermentation, although without linear evolution.

The in silico simulation results suggest that the inoculum load and the starting pH affect the acid production more than the fermentation temperature. From a practical point of view, the fermentation process could be more sustainable using a low inoculum load (4 or 5 log cfu/g), a temperature of 30 °C, and an overnight fermentation time (about 14–17 h), although leading to a Bio21B enriched in organic acids can be obtained.

In conclusion, this approach can be useful for scientists and food companies to optimize the starting fermentation conditions, depending on the needs, allowing them to reach the desired strain density, *pH*, and metabolite pattern in a shorter time. Further studies will be performed to test the liquid sourdough in the bread-making process and assess the nutritional features of the resulting product.

## Figures and Tables

**Figure 1 foods-11-03942-f001:**
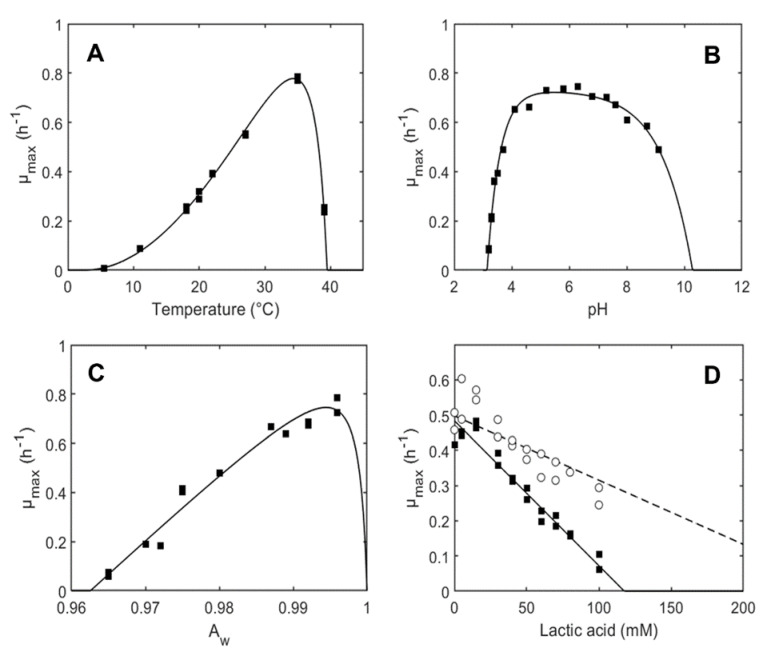
Effect of temperature (*T*) (**A**), pH (**B**), *a_w_* (**C**), and undissociated lactic acid [*HA*] concentration at pH 4.7 (straight line) and pH 5.1 (dashed line) (**D**) on the maximum specific growth rate (µ*_max_*) of the *L. plantarum* ITM21B strain. Comparison between the fitted model and observed maximum specific growth rates (▪,o).

**Figure 2 foods-11-03942-f002:**
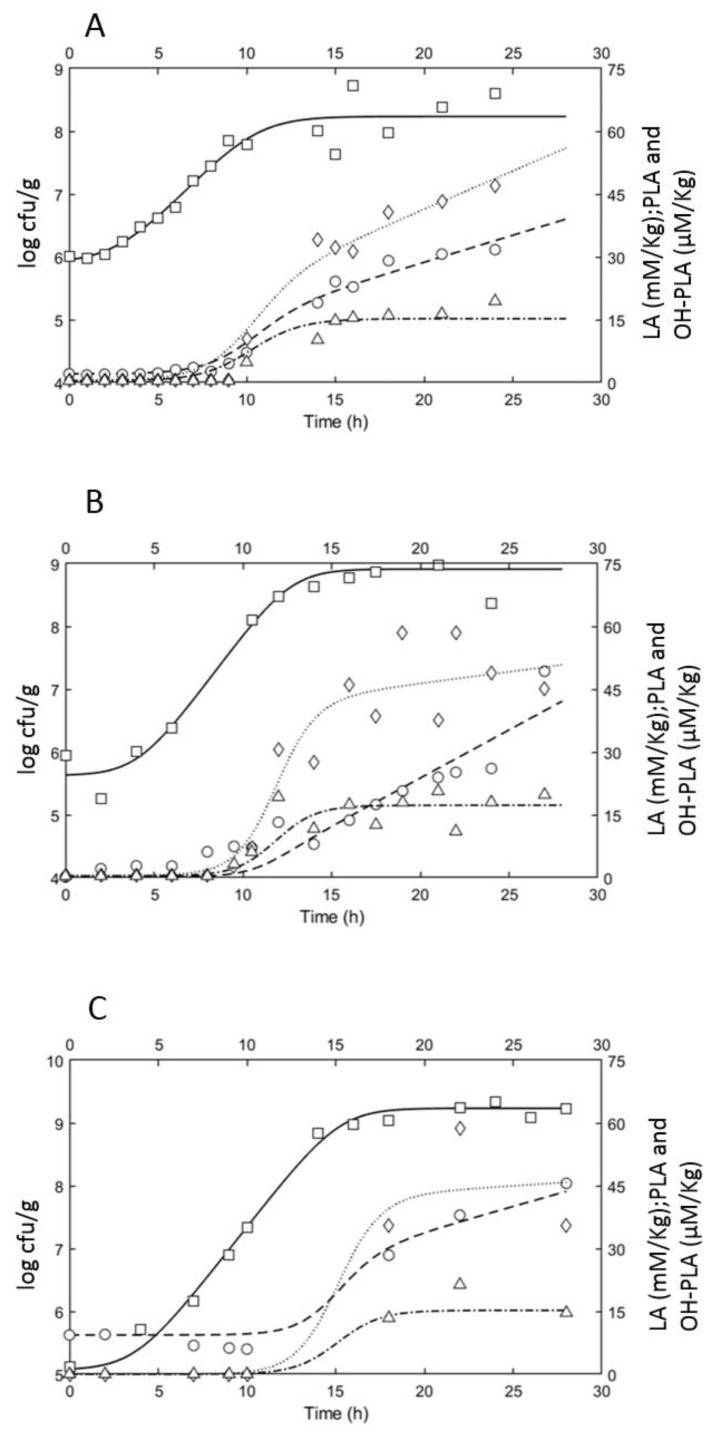
Comparison between the fitted kinetics of *L. plantarum* ITM21B growth and of organic acid production in liquid sourdough Bio21B-1 (**A**), Bio21B-2 (**B**), and Bio21B-3 (**C**), and experimental observations (*L. plantarum* log cfu/g: □, straight line; lactic acid (mM/kg) (LA): dashed line, ◯; phenyllactic acid (μM/kg) (PLA): dotted line, ◊; hydroxy-phenyllactic (μM/kg) (OH-PLA): dash-dotted line, Δ) during fermentation.

**Figure 3 foods-11-03942-f003:**
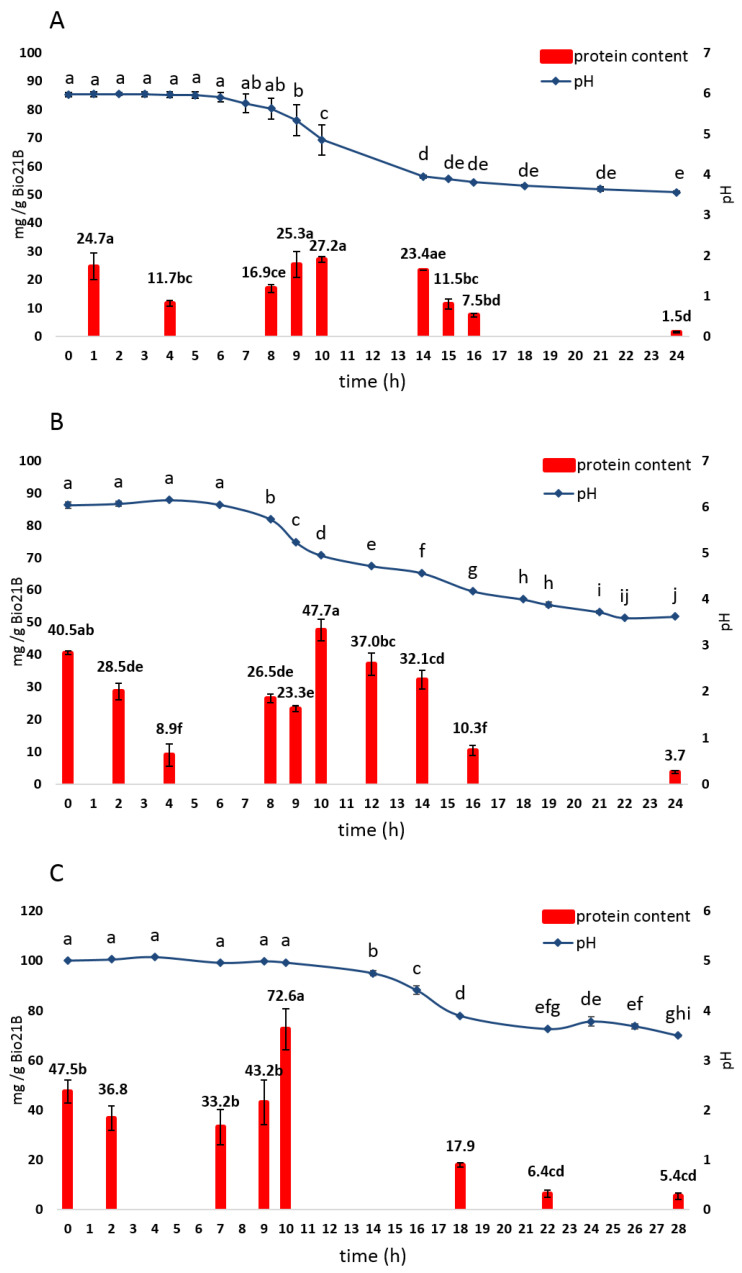
pH values and total protein content (mg/g) in liquid sourdough Bio21B-1 (**A**), Bio21B-2 (**B**), and Bio21B-3 (**C**) prepared as reported in Section 2.3 and Table 1, registered during fermentation. The values represent means of triplicates ± standard deviation. Different letters mean significant statistical differences (*p* < 0.05) among fermentation times within the same sample as determined by one-way ANOVA followed by Tukey’s HSD test.

**Figure 4 foods-11-03942-f004:**
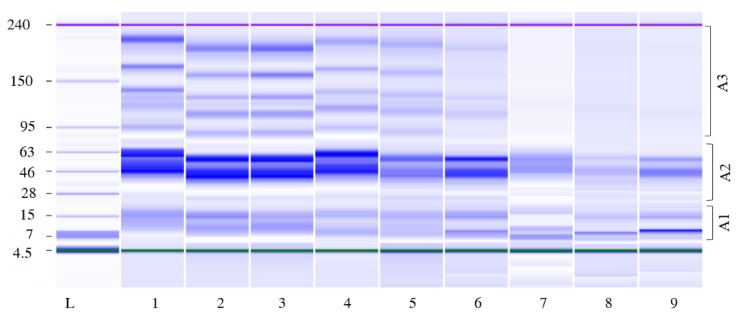
Electrophoretic analysis (LoaC) of total proteins extracted from liquid sourdough Bio21B samples during *L. plantarum* ITM21B fermentation shown as a gel-like image on Protein230 LabChip. Total proteins were extracted at fermentation times T0h (lanes 1, 2, and 3), T14h (lanes 4, 5, and 6), and T24h (lanes 7, 8, and 9) from Bio21B-1, Bio21B-2, and Bio21B-3, respectively; sizing Ladder (L). Numbered brackets indicate molecular weight areas: A1 (14–30 kDa), A2 (31–79 kDa), and A3 (80–230 kDa).

**Figure 5 foods-11-03942-f005:**
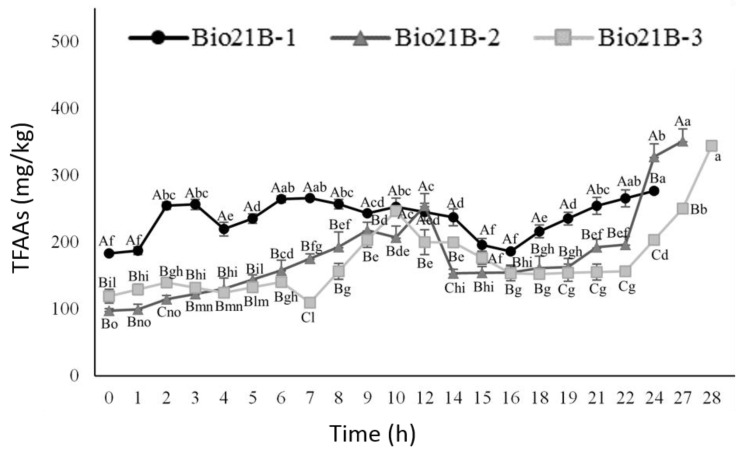
Content of TFAA—total free amino acid content (mg/kg) in liquid sourdoughs Bio21B-1, Bio21B-2, and Bio21B-3 prepared as reported in Section 2.3 and Table 1, during *L. plantarum* ITM21B fermentation. Data are the means of three replicates and bars indicate standard deviation. Different capital letters (A–C) indicate significant differences (*p* < 0.05) among different samples within the same incubation time. Different lowercase letters (a–o) indicate significant differences (*p* < 0.05) among incubation times within the same sample.

**Figure 6 foods-11-03942-f006:**
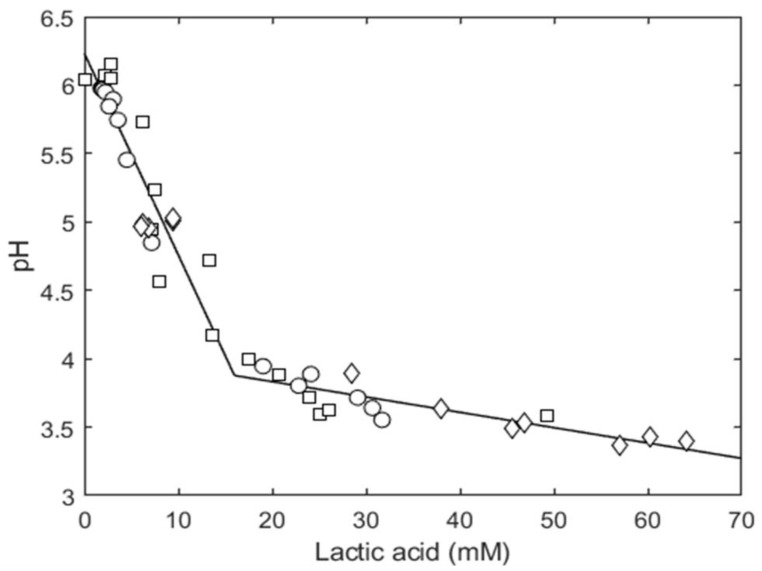
Evolution of pH of liquid sourdough as a function of the concentration of lactic acid. Comparison between experimental observations (◯: Bio21B-1, □: Bio21B-2, ◊: Bio-21B -3).

**Figure 7 foods-11-03942-f007:**
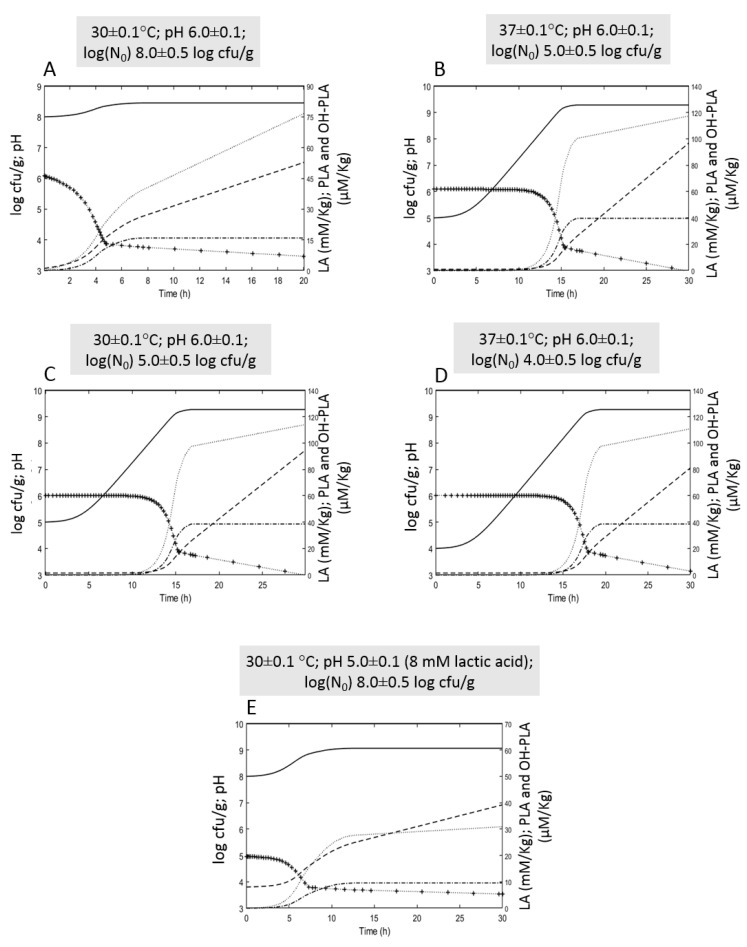
In silico growth simulations, pH reduction, and organic acid production in liquid sourdough at different starting fermentation conditions. Scenario (**A**): 30 °C, pH 6.0, log(N_0_) 8.0 cfu/g; Scenario (**B**): 37 °C, pH 6.0, log(N_0_) 5.0 cfu/g; Scenario (**C**): 30 °C, pH 6, log(N_0_) 5.0 cfu/g; Scenario (**D**): 37 °C, pH 6.0, log(N_0_) 4.0 cfu/g; Scenario (**E**): 30 °C, pH 5.0 (8 mM lactic acid), log(N_0_) 8.0 cfu/g. Log cfu/g: straight line; pH: plus-dotted line; lactic acid (LA): dashed line; phenyllactic acid (PLA): dotted line; hydroxy-phenyllactic (OH-PLA): dash-dotted line.

**Table 1 foods-11-03942-t001:** Starting fermentation conditions used to study the metabolite evolution and growth of *L. plantarum* ITM21B in liquid sourdough (Bio21B) at different inoculum load log(N_0_), temperature (T (°C), pH and lactic acid (LA) levels.

Starting Fermentation Conditions				
Liquid Sourdough	log(N_0_) ± SD	T (°C) ± SD	pH ± SD	LA (mM/Kg) ± SD
Bio21B-1	6.00 ± 0.32 a	30.0 ± 0.1 b	5.97 ± 0.06 a	2.06 ± 0.60 b
Bio21B-2	5.95 ± 0.00 a	37.0 ± 0.1 a	6.04 ± 0.07 a	<LOD b
Bio21B-3	5.11 ± 0.00 b	30.0 ± 0.1 b	5.00 ± 0.00 b	9.35 ± 4.36 a

The values represent means of triplicates ± standard deviation (SD). The liquid sourdough acronyms (Bio21B-1, Bio21B-2, Bio21B-3) are defined in the experimental section (Section 2.3) and differ for the starting fermentation conditions. Different letters in the same column mean significant statistical differences (*p* < 0.05) in the one-way ANOVA followed by Tukey’s HSD test. Limit of detection (LOD) is: 0.263 mM/kg.

**Table 2 foods-11-03942-t002:** Estimated growth cardinal parameters for temperature (*T*), *pH*, water activity (*a_w_*), and minimum inhibitory concentration of undissociated lactic acid (*MIC_U_*) and 95% confidence interval for the *L. plantarum* ITM21B in modified MRS.

	ITM21B Mean Value (Confidence Interval)
µ*_opt,MRS_* (h^−1^) for T	0.78 (0.74–0.81)
*T_min_* (°C)	2.40 (1.43–3.36)
*T_opt_* (°C)	34.35 (33.77–34.93)
*T_max_* (°C)	39.47 (39.32–39.62)
Number of data for T	14
R^2^	0.99
*pH_min_*	3.14 (3.13–3.16)
*pH_max_*	10.29 (9.54–11.03)
*Q*	0.41 (0.17–0.64)
Number of data for pH	19
R^2^	0.99
*a_w,min_*	0.963 (0.961–0.964)
*a_w,opt_*	0.994 (0.992–0.997)
Number of data for *a_w_*	13
R^2^	0.97
*MIC_U_ (mM)*	14.8 (14.0–15.7)
Number of data for [*HA*]	38 (19 at pH 4.7 and 19 at pH 5.1)

Estimated growth parameters (µ*_opt,MRS_*—optimum growth rate in MRS; *T_min_*—minimum growth temperature, *T_opt_*—optimum growth temperature, *T_max_*—maximum growth temperature, *pH_min_*—minimum growth *pH*, *pH_max_*—maximum growth pH_,_
*a_w,min_*—minimum growth water activity, *a_w,opt_*—optimum growth water activity), *MIC_U_*—minimum inhibitory concentration of undissociated lactic acid [*HA*] and the *Q*−Shape parameter for the pH model were calculated as reported in the experimental section (Section 2.7.2). R^2^—coefficient of determination.

**Table 3 foods-11-03942-t003:** Predicted kinetic parameters determined in liquid sourdough Bio21B-1, Bio21B-2, and Bio21B-3 using the model parameters for *L. plantarum* ITM21B and Equation (4).

Kinetic Growth Parameters of ITM21B in Liquid Sourdough				
	*lag* (h)	log (N_max_)	µ*_max_* (h^−1^)	µ*_max pred_* (h^−1^)
Bio21B-1	2.2	8.25	0.70	0.88
Bio21B-2	3.5	8.92	0.88	0.88
Bio21B-3	3.4	9.24	0.83	0.85

The liquid sourdoughs Bio21B were prepared as reported in Section 2.3 and differed for the starting fermentation conditions (Table 1); *lag*—initial phase of growth; log (N_max_)—maximum population density; µ*_max_* (h^−1^)—maximum growth rate of strain ITM21B in each liquid sourdough Bio21B calculated from Equation (2); *μ*_max pred_—maximum growth rate of strain ITM21B predicted in each liquid sourdough Bio21B calculated from Equation (9).

**Table 4 foods-11-03942-t004:** Estimated kinetic parameters *Y_p_* (mM/kg/cell.h or μM/kg/cell.h) and *m_p_* (mM/kg/cell.h or μM/kg/cell.h) relevant to lactic, phenyllactic (PLA), and hydroxy-phenyllactic (OH-PLA) acid production by *L. plantarum* ITM21B in the Bio21B samples.

		Estimated Kinetic Parameters	
Liquid Sourdough	Acid	*Y_P_* (mM/kg/cell.h or μM/kg/cell.h)	*m_p_* (mM/kg/cell.h or μM/kg/cell.h)
Bio21B-1	LA	8.24 × 10^−8^ (6.37 × 10^−8^–10.11 × 10^−8^)	7.49 × 10^−9^ (5.30 × 10^−9^–9.68 × 10^−9^)
	PLA	1.35 × 10^−7^ (9.92 × 10^−7^–1.70 × 10^−7^)	1.06 × 10^−8^ (0.64 × 10^−8^–1.48 × 10^−8^)
	OH-PLA	8.70 × 10^−8^ (7.66 × 10^−8^–9.73 × 10^−8^)	0
Bio21B-2	LA	6.08 × 10^−9^ (−4.43 × 10^−9^–16.60 × 10^−9^)	2.80 × 10^−9^ (1.65 × 10^−9^–3.96 × 10^−9^)
	PLA	5.06 × 10^−8^ (3.65 × 10^−8^–6.47 × 10^−8^)	6.57 × 10^−10^ (−8.82 × 10^−10^–21.96 × 10^−10^)
	OH-PLA	2.12 × 10^−8^ (1.78 × 10^−8^–2.45 × 10^−8^)	0
Bio21B-3	LA	1.10 × 10^−8^ (0.81 × 10^−8^–1.39 × 10^−8^)	7.05 × 10^−10^ (5.11 × 10^−10^–8.99 × 10^−10^)
	PLA	2.45 × 10^−8^ (1.63 × 10^−8^–3.26 × 10^−8^)	1.95 × 10^−10^ (−0.20 × 10^−10^–0.59 × 10^−10^)
	OH-PLA	1.00 × 10^−8^ (0.91 × 10^−8^–1.10 × 10^−8^)	0

*Y_p_*—growth-associated coefficient and *m_p_*—non-growth-associated coefficient were calculated as reported in paragraph 2.7.3. LA—Lactic acid; PLA—phenyllactic acid; OH-PLA—hydroxy-phenyllactic. Liquid sourdough Bio21B-1, Bio21B-2, and Bio21B-3 samples were prepared as reported in Section 2.3 and differed for the starting fermentation conditions (Table 1).

## Data Availability

The date are available from the corresponding author.

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
