# Peer review of "Modeling of Growth and Organic Acid Kinetics and Evolution of the Protein Profile and Amino Acid Content during Lactiplantibacillus plantarum ITM21B Fermentation in Liquid Sourdough"

_foods, 2022, doi:10.3390/foods11233942_

Round 1
Reviewer 1 Report
REVIEWER COMMENTS FOR MANUSCRIPT AFB (Beer as a vehicle for probiotics)
I critically reviewed the manuscript entitled “Modeling of Growth and Organic Acids Kinetics and Evolution of Protein Profile and Amino Acid Content During Lactiplanti- bacillus Plantarum ITM21B Fermentation in Liquid Sourdough”.
The authors collected recent data regarding criteria of biomass and acid production by few Lactobacillus genera in a very interesting manner. The article is well written and organized.
the major revision is made, it can be evaluated again regarding the quality of the manuscript. Several major revisions that need to be made by the author to improve this review manuscript include:
1- Please provide a goal and also practical conclusion, it could be helpful for any one.
2- Also point to the background and unsolved problem in the beginng of the abstract.
3- Whay Lactiplantibacillus Plantarum has been selected for study on its lactic (organic) acid production? While there are many other known probiotics in which acid production is higher.
4- If organic acid measurement is goal of this research why propionic acid with crucial health beneficial impacts has been ignored?
5- There is a discontinuity between the title of the article and the opening sentences of the introduction
6- I can not understand the meaning of the following sentence and its relation with fefore and after sentence:
This biotechnological approach exploits the metabolic features of microorganisms occurring in raw materials but can also be standardized using selected bacterial strains to pilot the fermentation
7- Please distribute references in exact place instead of listing 8 articles in the end of a general sentence:
LAB are intentionally added to start food fermentations [1], to confer functional properties to foods when used as probiotics [2‒7], to prolong food shelf-life when applied as bioprotective cultures or as producers of antimicrobial metabolites [8‒14].
8- Among LAB, the Lactiplantibacillus plantarum is consid- 40 ered the most representative species in sourdough and has been widely investigated as 41 starter since its proteolytic activity and ability to produce antimicrobial compounds [15] .
9- Gap of research has not been described in Introduction.
10- Please consider new paragraphing rule to avoid a mega paragraph and another one with just 2 sentence.
11- In the last paragraph of the introduction, novelty of this work should be mentioned and also define Who are the target community? Policy makers? Researchers? Sourdough Manufacturers?
12- - There is no track of Statistical calculation throught this text: Just for instance:
13- Is it a significant difference really between 9 and 5.99 in table 1? How much was P value?
14- I recommend adding seri of data which authors used to start their modeling. It couls be considered as supplementary or in text.
15- Provide a deep discussion via comparison with similar or controversial reports.
16- Please give a practical conclusion. I could not give the concept of "salt reduced yeast-leavened bread?!".
Author Response
Reviewer 1
REVIEWER COMMENTS FOR MANUSCRIPT AFB (Beer as a vehicle for probiotics)
I critically reviewed the manuscript entitled “Modeling of Growth and Organic Acids Kinetics and Evolution of Protein Profile and Amino Acid Content During Lactiplanti- bacillus Plantarum ITM21B Fermentation in Liquid Sourdough”.
The authors collected recent data regarding criteria of biomass and acid production by few Lactobacillus genera in a very interesting manner. The article is well written and organized.
the major revision is made, it can be evaluated again regarding the quality of the manuscript. Several major revisions that need to be made by the author to improve this review manuscript include:
Thank you to the reviewer for all valuable comments. Below are reported the responses to comments and suggestions
1- Please provide a goal and also practical conclusion, it could be helpful for any one.
The goal and practical conclusions have been added (lines 102-105; lines 683-696)
2- Also point to the background and unsolved problem in the beginng of the abstract.
The abstract has been accordingly modified (lines 15-19)
3- Whay Lactiplantibacillus Plantarum has been selected for study on its lactic (organic) acid production? While there are many other known probiotics in which acid production is higher.
As now more detailed in abstract and introduction sections (lines 20, 106-109), the L. plantarum 21B was chosen as sourdough starter strain able to pilot the fermentation process and produce organic acids with a role in the shelf-life extension of sourdough bread (Lavermicocca et al 2000) and in the taste of salt reduced bread, together with TFAA (Valerio et al. 2017).
4- If organic acid measurement is goal of this research why propionic acid with crucial health beneficial impacts has been ignored?
As detailed in the introduction section (lines 102-105), the goal of the manuscript was to use the mathematical modeling to characterize the growth and metabolism performances of the strain mainly in relation to its ability to produce a liquid sourdough that can be applied as salt replacer in bread since the content of organic acids lactic, PLA and OH-PLA, TFAA and proteins). The strain L. plantarum ITM21B used in the study is not a producer of propionic acid.
5- There is a discontinuity between the title of the article and the opening sentences of the introduction
Response: The introduction section has been critically reviewed accordingly to the Reviewer’s comments. (lines 35-43)
6- I can not understand the meaning of the following sentence and its relation with fefore and after sentence:
This biotechnological approach exploits the metabolic features of microorganisms occurring in raw materials but can also be standardized using selected bacterial strains to pilot the fermentation
Response: The sentence has been modified (lines 39-43)
7- Please distribute references in exact place instead of listing 8 articles in the end of a general sentence:
LAB are intentionally added to start food fermentations [1], to confer functional properties to foods when used as probiotics [2‒7], to prolong food shelf-life when applied as bioprotective cultures or as producers of antimicrobial metabolites [8‒14].
Response: the text has been accordingly modified (lines 44-49)
8- Among LAB, the Lactiplantibacillus plantarum is considered the most representative species in sourdough and has been widely investigated as starter since its proteolytic activity and ability to produce antimicrobial compounds [15].
Response: the text has been accordingly modified (lines 49-50)
9- Gap of research has not been described in Introduction.
Response: additional references have been introduced to cover all arguments treated in the Introduction section
10- Please consider new paragraphing rule to avoid a mega paragraph and another one with just 2 sentence.
Paragraphs in MM section relevant to the TFAA and protein content and profile have been combined as also those reporting the determination of kinetic growth parameters of L. plantarum ITM21B in Bio21B for the in silico simulations (line 218, 241, 359)
In the result section, the paragraphs have been divided to avoid big paragraph according to the reviewer’s request (lines 432, 467, 510, 572)
11- In the last paragraph of the introduction, novelty of this work should be mentioned and also define Who are the target community? Policy makers? Researchers? Sourdough Manufacturers?
Response: the required information has been added (lines 117-119)
12- - There is no track of Statistical calculation throught this text: Just for instance:
13- Is it a significant difference really between 9 and 5.99 in table 1? How much was P value?
The values reported in table 1 represent the starting fermentation conditions deliberately modified to study the changes occurring during fermentation in term of bacterial density, pH, TTA, metabolite production. As required, the statistical analysis (Anova) was performed within each column (same parameter) (lines 188-191; 384-385)
14- I recommend adding seri of data which authors used to start their modeling. It couls be considered as supplementary or in text.
The growth rate data vs. T, pH, aw and lactic acid have been made available in a supplementary file. (Supplementary tables 1 and 2)
15- Provide a deep discussion via comparison with similar or controversial reports.
As requested by the Reviewer, additional comments have been added throughout the text to discuss our data in comparison to the available published relevant papers.
16- Please give a practical conclusion. I could not give the concept of "salt reduced yeast-leavened bread?!".
A conclusion has been added as suggested by the reviewer (lines 670-696).

Reviewer 2 Report
Manuscript entitled “Modeling of Growth and Organic Acids Kinetics and Evolution of Protein Profile and Amino Acid Content During Lactiplantibacillus Plantarum ITM21B Fermentation in Liquid Sourdough”. In this paper, the cardinal growth parameters of the Lactiplantibacillus plantarum ITM21B strain were investigated. Overall, the approach and structure of article are integrity, and the analysis is relatively comprehensive. The paper has some issues, which should be addressed prior.
1、 The abstract should always be concise and informative. The arguments of why your study is important not making any sense. Extensive revision is required in abstract, as the present sentences sounds noisy during reading. Overall the abstract is not informative enough and need to show the actual picture of the work.
2、 Authors are requested to indicate the numerical values with significant difference in Table 1.
3、 Why the points number is different in subgraph of Fig 1? And please add the R2 for the fitted model.
4、 Line 451, after ca 22h?
5、 Please add the numerical values with significant difference in Fig 3. The current graph is hard in distinguish the pH values and total protein content, so the legends need to added.
6、 Which software was used to calculate the percentages of peaks for each Mw area?
Author Response
Reviewer 2
Manuscript entitled “Modeling of Growth and Organic Acids Kinetics and Evolution of Protein Profile and Amino Acid Content During Lactiplantibacillus Plantarum ITM21B Fermentation in Liquid Sourdough”. In this paper, the cardinal growth parameters of the Lactiplantibacillus plantarum ITM21B strain were investigated. Overall, the approach and structure of article are integrity, and the analysis is relatively comprehensive. The paper has some issues, which should be addressed prior.
1、 The abstract should always be concise and informative. The arguments of why your study is important not making any sense. Extensive revision is required in abstract, as the present sentences sounds noisy during reading. Overall the abstract is not informative enough and need to show the actual picture of the work.
Thank you for all your helpful suggestions. The abstract has been revised accordingly to the reviewer’s comments.
2、 Authors are requested to indicate the numerical values with significant difference in Table 1.
Table 1 has been accordingly modified
3、 Why the points number is different in subgraph of Fig 1? And please add the R2 for the fitted model.
The number of data points indicated in Table 1 were corrected and are now consistent with those in Fig 1.
4、 Line 451, after ca 22h?
The text has been corrected.
5、 Please add the numerical values with significant difference in Fig 3. The current graph is hard in distinguish the pH values and total protein content, so the legends need to added.
The Fig. 3 has been modified to make it clearer.
6、 Which software was used to calculate the percentages of peaks for each Mw area? LISA
Response: As reported in MM (line 266-267; 271-272), data evaluation was carried out by the dedicated 2100 Expert software. The software displays automatically data referred to each sample in a tabular format reporting for each peak information on molecular weight, relative concentration, peak area and the percentage respect to the totality of peaks. For each molecular weight area, the sum of these percentages was calculated.

Round 2
Reviewer 1 Report
Line 44-46: there are lots of good citation for LAB application in foods please cite to few articles to :
https://journals.sbmu.ac.ir/afb.
Figures are really in a very bad quality none of them are acceptable and Figure 3 is the worst. Please replace.
Don’t use abbreviation in Tables and Figures, without any citation in footnote of table.
As I am nor expert in the field of modelling, I recdommend Open review invitation after publication. In this case we can accept to release it and waiting for responses from readers.
If this is not applicable I strongly recommend sending email to expert in the field of modeling.
Best
Author Response
Response to Reviewer 1
Dear Reviewer 1, thank you for the deep revision on the manuscript. I hope that the revised version now submitted has been improved as you required. All changes have been marked in yellow color. All the comments addressed to authors have been considered as reported below.
Line 44-46: there are lots of good citation for LAB application in foods please cite to few articles to :
https://journals.sbmu.ac.ir/afb.
Response: as required by the Reviewer, few articles relevant to the LAB application in foods have been cited. Now the sentence (lines 45-47) has been simplified citing only four review articles including one from Appl. Microbiol. Biotechnol.
Figures are really in a very bad quality none of them are acceptable and Figure 3 is the worst. Please replace.
Response: I apologize for the inconvenience. Maybe the problem occurs since figures were included into the manuscript in the wrong way. Now all figures have been modified and included into the text as tiff images to improve their resolution.
Don’t use abbreviation in Tables and Figures, without any citation in footnote of table.
Response: all details on abbreviations used in tables and Figures have been included as indicated into the Instruction for Authors of the Journal.
As I am nor expert in the field of modelling, I recdommend Open review invitation after publication. In this case we can accept to release it and waiting for responses from readers.
If this is not applicable I strongly recommend sending email to expert in the field of modeling.
